



# Greenhouse Gas Retrievals for the CO2M mission using the FOCAL method: First Performance Estimates

Stefan Noël[1], Michael Buchwitz[1], Michael Hilker[1], Maximilian Reuter[1], Michael Weimer[1], Heinrich Bovensmann[1], John P. Burrows[1], Hartmut Bösch[1], and Ruediger Lang[2]

[1]Institute of Environmental Physics, University of Bremen, FB 1, P.O. Box 330440, 28334 Bremen, Germany
[2]EUMETSAT, Eumetsat Allee 1, 64295 Darmstadt, Germany

**Correspondence:** S. Noël (stefan.noel@iup.physik.uni-bremen.de)

**Abstract.** The Anthropogenic Carbon Dioxide Monitoring (CO2M) mission is a constellation of satellites currently planned to be launched in 2026. CO2M is planned to be a core component of a Monitoring and Verification Support (MVS) service capacity under development as part of the Copernicus Atmosphere Monitoring Service (CAMS). The CO2M radiance measurements will be used to retrieve column-averaged dry-air mole fractions of atmospheric carbon dioxide ($XCO_2$), methane

($XCH_4$) and total columns of nitrogen dioxide ($NO_2$). Using appropriate inverse modelling, the atmospheric greenhouse gas (GHG) observations will be used to derive United Nations Framework Convention on Climate Change (UNFCCC) COP 21 Paris Agreement relevant information on GHG sources and sinks. This challenging application requires highly accurate $XCO_2$ and $XCH_4$ retrievals. Three different retrieval algorithms to derive $XCO_2$ and $XCH_4$ are currently under development for the operational processing system at EUMETSAT. One of these algorithms uses the heritage of the FOCAL (Fast atmOspheric

traCe gAs retrievaL) method, which has already successfully been applied to measurements from other satellites. Here, we show recent results generated using the CO2M version of FOCAL, called FOCAL-CO2M.

To assess the quality of the FOCAL-CO2M retrievals, a large set of representative simulated radiance spectra has been generated using the radiative transfer model SCIATRAN. These simulations consider the planned viewing geometry of the $CO_2$ instrument and corresponding geophysical scene data (including different types of aerosols and varying surface properties)

which were taken from model data for the year 2015. We consider instrument noise and systematic errors due to the retrieval method but have not considered additional error sources due to e.g. instrumental issues, spectroscopy, or meteorology. On the other hand, we have also not taken advantage in this study of CO2M's MAP (Multi Angle Polarimeter) instrument, which will provide additional information on aerosols and cirrus clouds. By application of the FOCAL retrieval to these simulated data confidence is gained that the FOCAL method is able to fulfil the challenging requirements on systematic errors for the CO2M

mission (spatio-temporal bias $\leq 0.5\,\mathrm{ppm}$ for $XCO_2$ and $\leq 5\,\mathrm{ppb}$ for $XCH_4$).

## 1 Introduction

Carbon dioxide ($CO_2$) and methane ($CH_4$) are the two most important anthropogenic atmospheric greenhouse gases. Their atmospheric concentrations are rising as a result of anthropogenic activity. There is a scientific consensus that this is driving


global warming and related climate change (see the recent report of the Intergovernmental Panel on Climate Change (IPCC),
2023). In November 2015, the Paris Agreement of the United Nations Framework Convention on Climate Change (UNFCCC)
was adopted to limit global warming to well below 2°C (UNFCCC, 2015). However this treaty introduced a preferred limit of
1.5°C. As part of the Paris agreement, progress of emission reduction efforts is tracked on a regular basis. In this context, the
European Commission (EC), the European Space Agency (ESA), the European Centre for Medium-Range Weather Forecasts
(ECMWF), the European Organisation for the Exploitation of Meteorological Satellites (EUMETSAT), and international ex-
perts are developing an operational capacity for monitoring anthropogenic $CO_2$ emissions as a new $CO_2$ service under the EC's
Copernicus program (e.g. Janssens-Maenhout et al., 2020; Balsamo et al., 2021). A core component of this Monitoring and Ver-
ification Support (CO2MVS) capacity are satellite observations, in particular data from the European Anthropogenic Carbon
Dioxide Monitoring (CO2M) satellite mission (ESA, 2020; Lespinas et al., 2020; Sierk et al., 2021), which – with additional
instrumentation – builds on the heritage of the CarbonSat concept (Bovensmann et al., 2010; Velazco et al., 2011; Buchwitz
et al., 2013; Broquet et al., 2018) and the first retrievals of the column average dry-air mole fractions of $CO_2$ ($XCO_2$) and $CH_4$
($XCH_4$) retrieved using passive remote sensing observations in the Near Infrared (NIR) and shortwave infrared (SWIR) made
by the Scanning Imaging Absorption Spectrometer for Atmospheric Chartography (SCIAMACHY) on Envisat (Burrows et al.,
1995; Bovensmann et al., 1999; Buchwitz et al., 2005).

CO2M is planned to be a core component of a Monitoring and Verification Support (MVS) service capacity under devel-
opment as part of the Copernicus Atmosphere Monitoring Service (CAMS) (Janssens-Maenhout et al., 2020; Balsamo et al.,
2021; Hegglin et al., 2022). The CO2M mission will consist of a constellation of 2–3 satellites which will monitor globally
$XCO_2$ and $XCH_4$. The satellites will be placed in a sun-synchronous polar orbit at $735\,km$ altitude with an equator crossing
at about 11:30 local time in a descending node. The first CO2M satellite is planned to be launched in 2026. Each satellite has
a payload comprising three instruments:

– An imaging spectrometer (CO2I), which measures the welling radiance from the top of the atmosphere in wavelength
ranges having atmospheric absorption, which on mathematical inversion yield the total and tropospheric columns of
nitrogen dioxide ($NO_2$), column-averaged dry-air mole fractions of atmospheric carbon dioxide ($XCO_2$) and methane
($XCH_4$), SIF (solar induced fluorescence) and additional quantities such as the column-averaged dry-air mole fractions
of water vapour ($XH_2O$). The spatial resolution of CO2I ground scenes is about $2 \times 2\,km^2$.

– A multi-angle polarimeter (MAP), from which aerosol data products are retrieved. The spatial resolution of MAP is
about $4 \times 4\,km^2$.

– A cloud imager (CLIM), which measures the upwelling radiance in a selection of broad band spectral channels. The
spatial resolution for CLIM is higher than that of CO2I being about $0.4 \times 0.4\,km^2$.

A driving motivation for the selection of CO2M was the quantification of anthropogenic emissions of $CO_2$. However other
important objectives of the mission include the provision of knowledge about anthropogenic $CH_4$ emissions and on large-scale
natural $CO_2$ and $CH_4$ surface fluxes.





Three different retrieval algorithms to derive $XCO_2$ and $XCH_4$ are currently under development for the operational processing system at EUMETSAT. One of the foreseen operational CO2M algorithms is based on the FOCAL (Fast atmOspheric traCe gAs retrieval) method (Reuter et al., 2017a, b). The other two algorithms are RemoTAP (Remote sensing of Trace gas and

60 Aerosol Product Lu et al., 2022) and Fusional-P-UOL-FP based on the retrieval algorithm as described in Cogan et al. (2012). The requirements on data product quality for these algorithms are high (ESA, 2020), systematic errors (spatio-temporal bias) should not exceed $0.5\,\mathrm{ppm}$ for $XCO_2$ (about 0.12%) and $5\,\mathrm{ppb}$ for $XCH_4$ (about 0.28%). The corresponding maximum random errors for $XCO_2$ and $XCH_4$ are $0.7\,\mathrm{ppm}$ and $10\,\mathrm{ppb}$, respectively, for a specific scenario (solar zenith angle $50°$, surface albedo in NIR, SWIR-1 and SWIR-2: 0.2, 0.1, 0.05).

In this paper we show recent results generated using the current CO2M version of FOCAL, which was applied to a set of simulated measurement data in order to assess the quality of the retrieved $XCO_2$ and $XCH_4$ data products. Special emphasis is placed on the verification of the systematic error requirements, which are actually more challenging for the retrieval. This is because the random errors are mainly related to the noise of the spectra which is determined by instrument design. Although the results are obtained from the analysis of top of the atmosphere radiances simulated using a state of the art radiative model,

this study provides some first estimates of the data product quality from the FOCAL-CO2M retrieval algorithm.

The structure of the paper is described and summarised as follows. After this introduction, we describe the input data used in this study and how they were generated in Sect. 2. In Sect. 3 we explain the FOCAL retrieval and the methods used for performance assessment. The results of the study are presented in Sect. 4. Finally, our conclusions are summarised in Sect. 5.

## 2 Input Data

The main input data used in this study are simulated radiance spectra in the near-infared (NIR) and short-wave infrared (SWIR) bands to be measured by CO2I (see Table 1). These have been generated using the SCIATRAN radiative transfer model (Rozanov et al., 2017) using CO2M geolocation and viewing geometry information provided by EUMETSAT as input. The SCIATRAN calculations are more complex than the FOCAL forward model. For example, they consider surface BRDF (bidirectional reflectance distribution function) effects, different aerosol types and distributions as well as clouds.

In the context of the current study we have generated two types of test data sets, which will be used for the performance assessments: (i) a full-year global data set with a reduced spatial sampling and (ii) a spatially high-resolved scene over Europe (the so-called 'Berlin scene'). Both are described in the following sub-sections.

In order to be as consistent as possible with real measurements, random noise has been added to the simulated spectra. This noise $N$ has been calculated for each radiance $R$ using band-specific parameters $A$ and $B$ via:

$$N = \sqrt{R\,A + B^2}/A \tag{1}$$

The assumed values for $A$ and $B$ are given in Table 2. These values were derived from a study on CO2M requirements and performance (Buchwitz et al., 2020) and have been shown to be consistent with the measurements of the selected CO2M detectors.



## 2.1 Full-year global subset

To assess the impact of large scale temporal and spatial variations on the FOCAL-CO2M results a global data set covering
      at least a full year is required. However, SCIATRAN simulations are computationally expensive. Therefore it is not currently
      feasible to compute a complete CO2I full year data set for one of the CO2M satellites within a reasonable time. For the
      full-year data we therefore selected a subset of CO2I measurement geometries containing every $15^{th}$ out of 110 across-track
      ground pixels and every $20^{th}$ out of roughly 9200 along-track scanlines per orbit for solar zenith angles lower than $80°$. This
results in a subset with 300 times less data than the whole CO2M data set, but with similar spatial and temporal coverage. The
      meteorological information (pressure, temperature, water vapour) used in the SCIATRAN simulations is taken from the fifth
      generation of the ECMWF re-analysis (ERA5) data (Hersbach et al., 2020) (temporal resolution $1\,h$, spatial resolution $0.25°$).
      $CO_2$ and $CH_4$ profiles use the results from the CAMS model data for 2015 (spatial resolution about $2° \times 3°$), namely v20r1
      for $CO_2$ (Chevallier et al., 2005; Chevallier et al., 2010; Chevallier, 2013) and v20r1 for $CH_4$ (Segers, 2022). The reflectiv-
ity of the surface is modelled using BRDF parameters from the Moderate Resolution Imaging Spectroradiometer (MODIS)
      MCD43C1 Version 6.1 BRDF and albedo model parameters dataset (Schaaf and Wang, 2021). Solar induced chlorophyll fluo-
      rescence irradiance is simulated by scaling an irradiance spectrum obtained from the publication of Rascher et al. (2009). The
      scaling factor is obtained by assuming a linear relationship between SIF irradiance at $740\,nm$ and MODIS NDVI derived by
      the Rutherford Appleton Lab (RAL, 2022). Clouds in the dataset are considered by SCIATRAN using ERA5 specific cloud
liquid water content and specific cloud ice water content as input. Aerosols are taken from the CAMS aerosol climatology
      (Inness et al., 2019). Topographic information is taken from GTOPO30 (Earth Resources Observation and Science Center,
      U.S. Geological Survey, U.S. Department of the Interior, 1997). The SCIATRAN calculations have been performed in scalar
      mode without consideration of inelastic scattering processes. We also assumed a uniform scene within each ground pixel. So
      far, only nadir data over land are modelled, which results in a total of about 6 million spectra per year for each band.
Fig. 1 shows as an example for the sampling of the $XCO_2$ subset data over part of Europe for one CO2M orbit (only data
      over land). The shown region corresponds to the range of the high-resolution scene addressed in the following sub-section.

## 2.2 High-resolution scene

      In addition to the full-year global subset data we used SCIATRAN to model also the NIR and SWIR radiances for a full 3
      minute granule of CO2I data containing about 67000 measurements of which about 37000 are over land and cloud-free. This
granule from 3 July 2015 (referred to as the 'Berlin scene', see Fig. 2) is one of the typical test scenes, used within the CO2M
      project, because of the availability of high-resolution model data for this scene. The calculations for the high-resolution scene
      use the same SCIATRAN setup except for geolocation, geopotential, pressure, temperature, specific humidity, $CO_2$ and $CH_4$,
      which were provided by EUMETSAT using high-spatial resolution ($9\,km$) data from the CAMS nature run model (Agustí-
      Panareda et al., 2022). As can be seen from Fig. 2, with this resolution $XCO_2$ plumes from power plants in Eastern Germany
are clearly visible, although the increase of $XCO_2$ in these plumes is only a few ppm above background. Fig. 3 shows the
      corresponding $XCH_4$ data.



## 3 Algorithms

### 3.1 FOCAL-CO2M retrieval

The FOCAL retrieval method is based on optimal estimation. FOCAL models the propagation of light through the atmosphere.
Scattering is approximated by a thin single scattering layer, which is characterised by the layer height (pressure level), the optical thickness of the layer and the Ångström coefficient describing the wavelength dependence of the scattering (see e.g. Reuter et al., 2017b, for details). Applications to OCO-2, GOSAT and GOSAT-2 have shown that FOCAL is fast and produces accurate results. For example, the spatio-temporal bias of the FOCAL $XCO_2$ product derived from TCCON comparisons is (after bias correction) in the order of 0.6 ppm for OCO-2 (Reuter and Hilker, 2022), and 0.6 (1.1) ppm for GOSAT (GOSAT-2) (Noël et al., 2022). FOCAL is therefore well suited for the analysis of large data sets.

FOCAL-CO2M is an adaptation of the FOCAL method for use in the CO2M mission. FOCAL permits "full physics" (FP) and "proxy" (PR) retrievals. FP retrievals are based on directly retrieving the quantity of interest, i.e., $XCO_2$ or $XCH_4$, whereas PR retrievals are based on computing the ratio of the retrievals of the two gases and using modelled $XCO_2$ or $XCH_4$ for correction (see, e.g. Schepers et al., 2012, for details). The main output products of the FOCAL-CO2M retrieval are total column FP $XCO_2$ and $XCH_4$, but there will also be corresponding additional PR data and SIF and water vapour ($XH_2O$) products. However, in the current study we only consider the FP $XCO_2$ and $XCH_4$ products.

The retrieval consists of three steps: pre-processing, inversion and post-processing.

The inputs to the pre-processing include (1) the spectral data from CO2I (measured radiances and their uncertainties as well as related measurement times and measurement geometry, geolocation etc.) and (2) related meteorological information and a-priori profiles for the considered gases. For the current study, we use simulated data, see Sect. 2.

The objective of pre-processing is to filter the input data to minimise the waste of computational time for unsuitable atmosphere and ground scenes or soundings. For the purpose of this study, we filter out all cloudy data and data over water surfaces, because these are not required for the verification of the systematic error requirements. However, the retrieval is also planned to be applied to data over ocean, especially in glint mode. We also remove all data with solar zenith angles larger than 75° and data for which the signal-to-noise is lower than 100 at the wavelengths, 755 nm, 1624 nm and 2036 nm, i.e. one spectral region in each band where absorption is low.

The inversion uses an optimal estimation retrieval approach (Rodgers, 2000). It has four fitting windows in the near-infrared (NIR) and short-wave infrared (SWIR) spectral regions, see Table 3. The corresponding state vector elements and their a-priori values are listed in Table 4. The assumed a-priori uncertainties for the gas profiles consider covariances and are the same as those used in GOSAT and GOSAT-2 FOCAL retrievals, which were derived based on the SLIM (Simple cLImatological Model for atmospheric $CO_2$ or $CH_4$) climatology (see Noël et al., 2022, for details). As for OCO-2, GOSAT and GOSAT-2, we consider an additional forward model error in the retrieval, which takes into account possible limitations of the forward model and is determined from the simulated CO2M measurements, see e.g. Reuter et al. (2017a, b) for details. The instrument line shape (ILS) functions are currently assumed to be Gaussian with a full-width at half maximum (FWHM) as given by the spectral resolution in Table 1.





During post-processing, the output data from the inversion are filtered for e.g. outliers. Furthermore, a bias correction is performed to remove systematic offsets arising e.g. from limitations of the forward model. The underlying data base for the post-processing is generated using a subset of (uncorrected) retrieval results as input. Here, we use the results of the retrieval after inversion for the April 2015 subset data. These data are only filtered for convergence and fit quality. Using these data, the current post-processing data base has been derived as follows.

The filtering of the data is similar to the filtering performed for OCO-2 and GOSAT(-2) data and comprises two filtering steps (see e.g. Noël et al., 2022, for details). First, data are filtered for retrieval quality (see Reuter et al., 2017b). Second, additional filter parameters and their limits are determined using a variance minimisation method. The idea of this second step is that outliers largely contribute to the, scatter and the method finds thresholds for parameters which most efficiently remove these outliers from the final data set.

In an iterative procedure, a set of maximal 10 parameters is determined which have the largest effect on the variance reduction of the bias, i.e. the difference between the retrieved value and an assumed 'true' value, for a prescribed percentage of data to be filtered out. The percentage of data to be filtered out is a trade-off between the remaining scatter of the data and the number of remaining data after filtering. For the simulated data used in the present study we prescribed that 15% shall be removed.

The bias correction is based on a machine learning regression, which determines the function and the 10 (or less) best parameters to reduce the bias based on a set of training and test data (each 50% of the input data). This is similar to the method described in Noël et al. (2022), but here we use a regression based on a gradient boosting method (currently XGBoost, Chen and Guestrin (2016)) instead of a random forest regression. For the current test data set, XGboost performs better than random forest regression.

The final determination of the post-processing data bases is done in an iterative way:

1. Determine and apply the bias correction to the unfiltered and uncorrected retrieval results.

2. Determine the (final) filter settings using the data from step 1 as input.

3. Filter the original (uncorrected / unfiltered) input data using these filters.

4. Determine the (final) bias correction based on the filtered (but uncorrected) data.

Performing a preliminary bias correction before the determination of the filter settings has the advantage, that data which can be sufficiently well corrected via the bias correction are not necessarily filtered out.

For simulated data, the 'true' $XCO_2$ and $XCH_4$ values are perfectly known, because they have been used for the generation of the simulated spectra. Therefore, the current filtering and bias correction does not consider any additional errors resulting from systematic differences between the estimated meteorological conditions and the actual atmosphere.

Note that the quality of the post-processing may in principle be improved by extending the input data set used to determine the post-processing data base (especially regarding the training of the bias correction). However, at the beginning of the CO2M mission, the amount of available measurement data will be limited. In order to show that our post-processing would even work with a minimum amount of data we use only one month of simulated data here.





## 3.2 Adaptations for real data

The FOCAL-CO2M retrieval software has been designed such that it can be applied to both simulated data (as in the present study) and to real measurement data. However, the application to actual measurements requires some adaptations.

This includes the incorporation of results from the on-ground calibration (e.g. updated ILS data) as well as updates of filtering and bias correction parameters, which can only be determined during the commissioning phase based on the analysis of in-flight measurements.

In the pre-processing, e.g. cloud and signal-to-noise filters need to be adjusted. For the retrieval, the forward model error needs to be re-determined. Furthermore, the post-processing data base needs to be re-calculated using adapted filter settings and bias correction parameters. It also has to be checked if additional information, e.g. aerosol parameters derived by the MAP instrument, may be used in both pre- and post-processing.

## 3.3 Performance Assessments

The primary objective of this study is to obtain a first estimate of the performance of the FOCAL-CO2M retrieval with respect to known sources of systematic errors. As already mentioned above, the corresponding requirements on the resulting $XCO_2$ and $XCH_4$ are high (systematic error $\leq 0.5\,\mathrm{ppm}$ and $5\,\mathrm{ppb}$, respectively), see ESA (2020).

However, these requirements are formulated in a general way. Therefore, there is a need for some interpretation to verify these requirements. We consider that the requirements should be verified by using cloud-free data over land only. Furthermore,
the interpretation of systematic errors depends on the application, i.e. the purpose for which the data shall be used. CO2M has two main application areas:

1. Quantification of anthropogenic emissions.

2. Quantification of natural large scale fluxes.

For the quantification of anthropogenic emissions it is important that local enhancements (e.g. emission plumes) can be
separated from the background. The background values themselves are less important. A verification therefore requires spatially highly resolved scene data with e.g. emission plumes.

For the quantification of natural large scale fluxes local variations are less relevant. Here, it is important that large scale structures and their variations in both time and space are correct. This requires global data covering at least a full year to consider possible long-term / large-scale errors.

According to the ESA (2020), $CO_2$ plume imaging (i.e. anthropogenic emissions) is the driving application for the precision requirements. Nevertheless, knowledge about larger scale or areal fluxes are also important for e.g. global modelling. Therefore, we consider both applications here. The verification of the requirements thus has to take these different scales into account. In the following sub-section we describe the verification methods for both application areas.

For the verification of the systematic error requirements for natural large scale fluxes we use the retrieval results from the
full-year global subset measurements as these provide a good spatial and temporal coverage. We then determine a running





average of the difference between the retrieved value and the true value within a $1° \times 1°$ latitude/longitude box. This results in a low-pass filtered bias data set. For this data set we compute the standard deviation, considering the cosine of the latitude as weights to account for different sizes of the averaging area. To fulfil the systematic error requirement, the resulting weighted standard deviation of the low-pass filtered bias should then be $\leq 0.5$ ppm.

For the verification of the systematic error requirements for anthropogenic emissions we take as input the high-resolution Berlin scene. We then apply – similar as for the large scale fluxes – a $1° \times 1°$ low-pass filter to the difference between the retrieved value and the true value, which results in a spatially smoothed bias. This smoothed bias is then subtracted from the original data, which gives us a high-passed filtered bias data set (for this scene). The standard deviation of these high-pass filtered bias data should then be $\leq 0.5$ ppm to fulfil the requirement on systematic errors.

Fig. 4 shows as an example for the different filtering procedures the unfiltered $XCO_2$ bias (retrieved - true values) for the high-resolution Berlin scene and the resulting low- and high-pass filtered bias.

## 4 Results

### 4.1 Application to anthropogenic emissions

As explained above, the verification of the performance requirements for anthropogenic emissions is achieved by using the 235 high-resolution 'Berlin scene'.

    Fig. 5 shows the FOCAL-CO2M $XCO_2$ retrieval results for this scene. The retrieved $XCO_2$ (after post-processing) is shown in the left plot, the true $XCO_2$ in the centre and their difference in the right plot. Some statistical information is also given in the figure below these plots.

    All structures of the scene shown in the true $XCO_2$ can also be identified in the retrieved data. The mean difference between 240 the retrieved and the true $XCO_2$ is -0.2 ppm. The standard deviation of the difference is 0.6 ppm. After application of the low- and high-pass filters this reduces to 0.2 ppm and 0.5 ppm. It should be noted that except for the low-pass standard deviation these values also include the noise on the data. The mean noise error of the data in this scene is also 0.5 ppm. This means that the high-pass standard deviation is dominated by noise; thus the real systematic error is probably well below 0.5 ppm. If we subtract the noise related variance from the high-pass variance and then take the square root, we get the value given in brackets 245 after the high-pass variance in the plot, namely 0.1 ppm. This can be considered as a lower estimate for the high-pass standard deviation as it does not consider potential systematic error contributions to the a-posteriori noise error.

    A small gradient is visible in the difference between the retrieved and the true $XCO_2$ from north-east to south-west. This could be related to aerosol effects. In this scene, most aerosol is located in the south-west. Differences in the handling of surface properties by SCIATRAN and FOCAL could also play a role. However, since we are interested in the quantification 250 for anthropogenic emissions, these larger scale effects are less relevant.

    As mentioned in Table 4, the $XCO_2$ a-priori values used in the retrieval are taken from the meteorological data, which where also used in the generation of the input spectra. They are therefore identical with the true values. This is because we aim to be as consistent as possible with the procedures to be applied to real data at a later time, and for real measurements we will





also use the (predicted) meteorological input data as truth for the post-processing corrections. However, in reality of course the truth will deviate from the model data. To show that the retrieval is not much sensitive to the choice of the a-priori, we have performed the retrieval for the 'Berlin scene' also for a fixed $CO_2$ a-priori for all measurements by assuming a constant value of 400 ppm for all altitudes. As can be seen from Fig. 6, this has hardly an impact on the retrieval results. All $XCO_2$ features can be reproduced even with the fixed a-priori. There is only a small mean offset of about 0.2 ppm compared to the values where the true $XCO_2$ was used as a-priori.

Fig. 7 shows a zoom-in of Fig. 5 on the region of the power plants in Eastern Germany. Despite the noise on the data, the plumes from the different power plants can be clearly identified in the retrieval results. No plume structures are visible in the difference map. The high-pass standard deviation for this sub-scene is 0.5 ppm similar to the noise error. The high-pass standard deviation is thus also dominated by noise. Subtraction of the noise contribution results in a standard deviation of 0.1 ppm. The requirement on systematic errors is therefore fulfilled for $XCO_2$ for this scene.

The results for $XCH_4$ using the true values as a-priori are shown in Fig. 8. For this case, also the main structures of the true $XCH_4$ field are re-produced in the retrieval. As for $XCO_2$, the difference between retrieved and true $XCH_4$ is dominated by noise. The mean offset for this scene is -1.4 ppb with a standard deviation of the (unfiltered) difference of 5.2 ppb, including a noise error of 4.8 ppb. The high-pass filtered mean standard deviation is 5.0 ppb with a lower (noise-corrected) estimate of 1.3 ppb. The requirement of maximum 5 ppb is therefore fulfilled. The difference map for $XCH_4$ shows a similar gradient as for $XCO_2$ from north-east to south west, most likely for the same reasons.

## 4.2 Application to large scale fluxes

The verification of the requirement for large scale natural fluxes is based on the full-year global subset data. As an example, Fig. 9 shows the FOCAL-CO2M $XCO_2$ retrieval results for April 2015. The corresponding $XCH_4$ data are shown in Fig. 10.

As can be seen from these figures, the retrieved data reproduce all large-scale patterns present in the true/a-priori data. The scatter in the differences between retrieved and true values is dominated by noise. Since the April 2015 data are used in the derivation of the post-processing data bases, Figs. 9 and 10 show the best case. However, the quantitative assessments described in the following show that other months have a similar performance.

For these quantitative assessments of the systematic error for large scale fluxes we apply a low-pass filter to the differences and determine the weighted standard deviation for the low-pass filtered data as described in section 3.3. This is done for each month as well as for the whole year 2015. The results are shown in Fig. 11 and Fig. 12.

For the verification of the systematic error requirements the red line in the middle plots is relevant. It shows the weighted standard deviation for the low-pass filtered data for each month and the value for the complete year (i.e. not the average over the monthly data) in the legend of each panel.

The yearly average low-pass standard deviation for $XCO_2$ is 0.5 ppm and therefore just fulfils the systematic error requirement for natural large scale fluxes. For $XCH_4$, the yearly average low-pass standard deviation is 3.7 ppb and therefore smaller than the required 5 ppb.





The lowest $XCO_2$ and $XCH_4$ standard deviations are achieved in April 2015. The biases at this month are also zero. This is not surprising, because this is the month which was used for the training of the bias correction. Slightly higher standard deviations occur in other months, but the standard deviations of the low-pass filtered data are always below $0.6\,ppm$ for $XCO_2$ and $4.\,ppb$ for $XCH_4$. Largest standard deviations occur for both gases in December and January, where also the number of valid data is lowest. In general, it is expected that the results improve if more months (e.g. a full year subset data set)are used for the generation of the post-processing data base.

### 4.3 Aerosol dependence

Up to the present, the FOCAL-CO2M retrieval does not use any external information about aerosols (e.g. from the MAP instrument). The systematic error requirements are only applicable up to an aerosol optical depth of 0.5 (ESA, 2020). Therefore, we also checked the aerosol dependence of the FOCAL-CO2M retrieval results. Figs. 13 and 14 show the (binned) differences between the retrieved and the true $XCO_2$ and $XCH_4$ for the full-year 2015 subset data as a function of the aerosol optical depth (AOD) at $550\,nm$, which was assumed for the generation of the simulated spectra with SCIATRAN.

As mentioned above, the SCIATRAN calculations for the simulated data consider different aerosol types and distributions, but the FOCAL retrieval does not explicitly consider aerosol, it assumes only one effective scattering layer. Nevertheless, as can be seen from Fig. 13, mean systematic offsets due to aerosol for the complete year are less than about $0.2\,ppm$ for $XCO_2$ with a mean of zero for all AODs up to 0.5. The standard deviation of the $XCO_2$ difference is on average $0.7\,ppm$ and typically smaller for lower AOD. The functional dependence on AOD is similar for $XCH_4$ (see Fig. 14). Systematic $XCH_4$ offsets are usually smaller than $1\,ppb$.

These results could possibly be improved when using extended training data for the post-processing and/or additional information from the MAP instrument.

### 5 Conclusions

FOCAL is one of three retrieval algorithms under development for the operational retrieval of $XCO_2$, $XCH_4$ and other parameters from the constellation of CO2M satellites to be launched from 2026 onward. These data products contain information on anthropogenic and natural sources and sinks of the two greenhouse gases $CO_2$ and $CH_4$ which will extracted using appropriate inverse modelling to support emission monitoring in the context of the Paris Agreement on climate change. This application requires high accuracy as even small biases can lead to significant emission errors (ESA, 2020).

The FOCAL retrieval has been successfully adapted to simulated CO2M data. First performance tests using data simulated with SCIATRAN as input have been performed. Based on cloud-free nadir data over land, we show that the requirement of a maximum systematic error of $0.5\,ppm$ for $XCO_2$ and $5\,ppb$ for $XCH_4$ is fulfilled by the FOCAL retrieval for (1) anthropogenic emissions (high-pass filtered data), using a high-resolution scene containing $XCO_2$ emission plumes from power plants and (2) natural large scale fluxes (low-pass filtered data), based on a full year global sub-sampled data set. Good retrieval results are obtained up to AOD 0.5, even without using external aerosol information as input.



All results shown here are based on simulated data. Furthermore, the calculations currently assume a perfect CO2I instrument
and do not consider any systematic errors in spectroscopy or meteorology. Information on aerosols and cirrus clouds derived
from the MAP instrument is also not considered yet.

Finally, the performance of the FOCAL-CO2M retrieval (and all other retrieval methods) needs to be determined based on
real measurements. Fulfilling the requirements for $XCO_2$ natural large scale fluxes is probably the most challenging task in
this context. However, the current results give good confidence that the FOCAL-CO2M retrieval is able to fulfil the product
quality requirements of the CO2M mission.

*Data availability.* The data used in this study are available on request from the authors.

*Author contributions.* R. Lang provided the CO2M geolocation information. M. Reuter developed the FOCAL method and provided the
geophysical input data for the SCIATRAN simulations. M. Hilker developed the original python implementation of FOCAL (OCO-2 version)
and computed the simulated spectra with SCIATRAN. S. Noël adapted the FOCAL method to CO2M and performed the retrievals and the
330 performance checks. All authors provided support in writing the paper.

*Competing interests.* The authors declare that there are no competing interests.

*Acknowledgements.* ERA5 meteorological data were provided by the European Center for Medium Range Weather Forecasts (ECMWF).

Large parts of the calculations reported here were performed on HPC facilities of the IUP, University of Bremen, funded under DFG/FUGG
grant INST 144/379-1 and INST 144/493-1.
The work has been carried out with funding by the European Union Copernicus program through EUMETSAT contract EUM/CO/19/4600002372/RL,
and the State and the University of Bremen.

Part of this work is funded by the BMBF project 'Integrated Greenhouse Gas Monitoring System for Germany - Observations (ITMS B)'
under grant number 01 LK2103A.



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



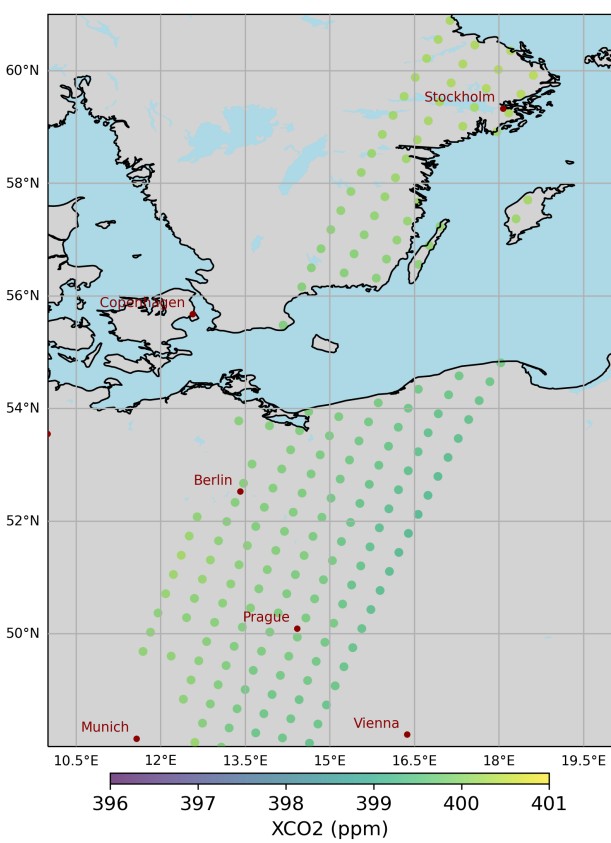

**Figure 1.** Example for the subset data: Modelled $XCO_2$ over part of Europe for one orbit on 2015-07-03. Only cloud-free data over land are shown because only these will be used later in the retrieval. No post-processing filters are applied. Note that only the centre points of the ground pixels are plotted and that the size of the markers is much larger than the original ground pixel size.



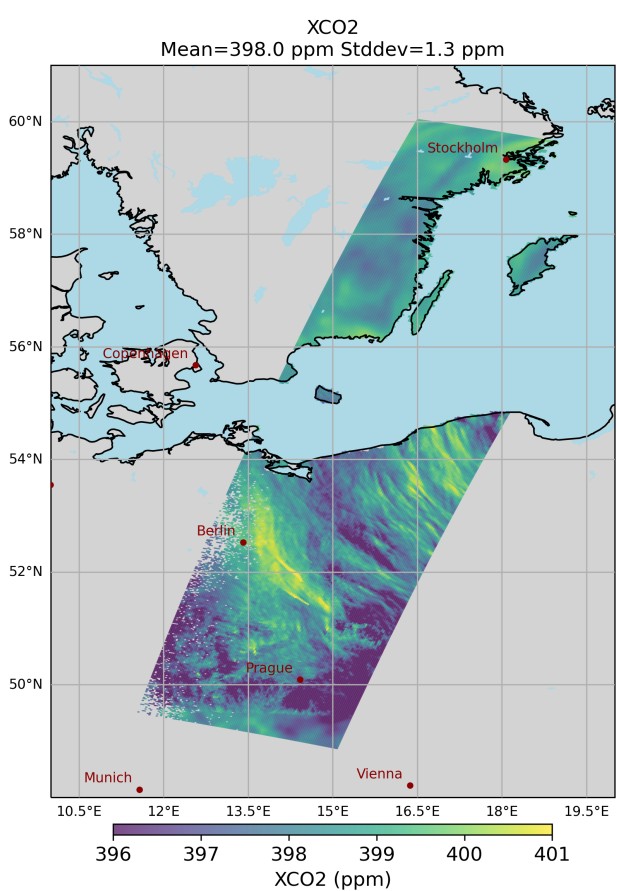

**Figure 2.** High-resolution scene: $XCO_2$. Only cloud-free data over land; no post-processing filters are applied.



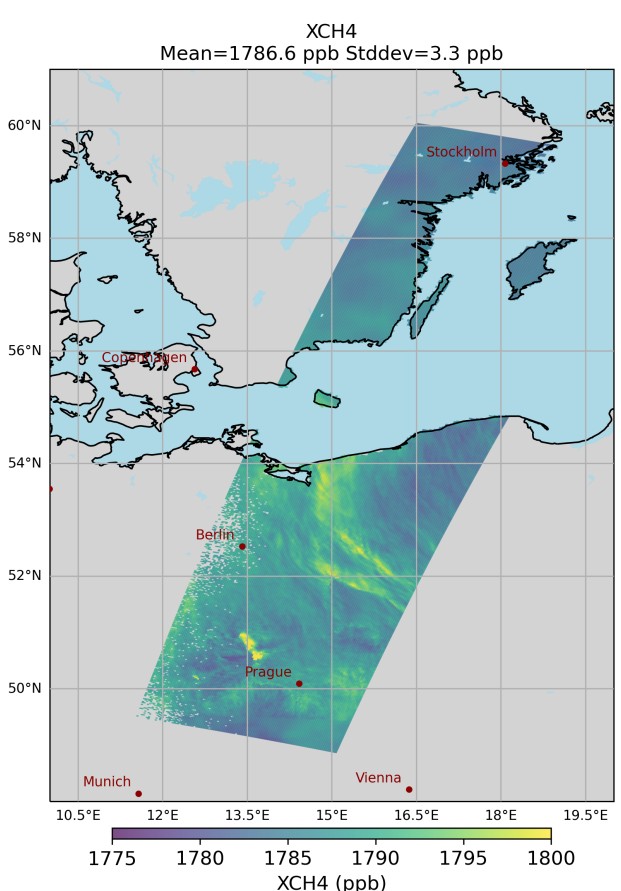

**Figure 3.** As Fig. 2, but for $XCH_4$.

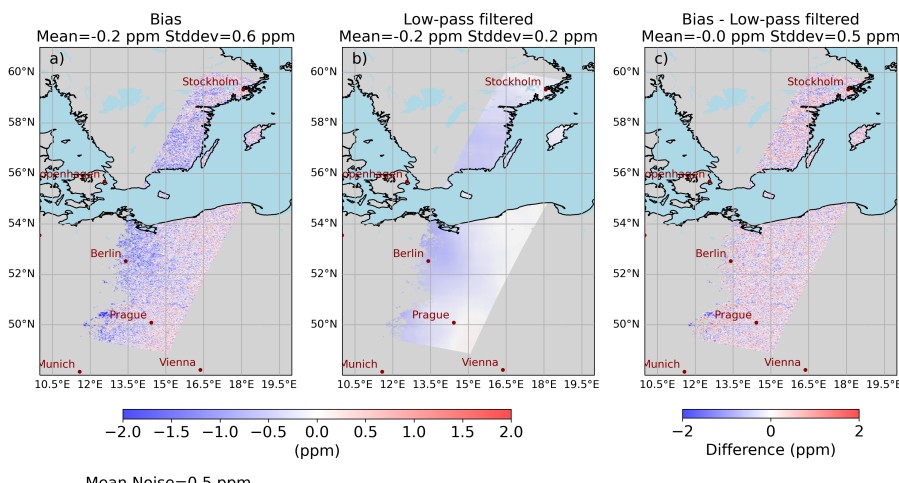

**Figure 4.** Example for low-/high-pass filtering (Berlin scene). a) Bias (FOCAL-CO2M retrieved - true $XCO_2$). b) $1° \times 1°$ low-pass filtered bias. c) High-pass filtered bias = Bias - Low-pass filtered bias.

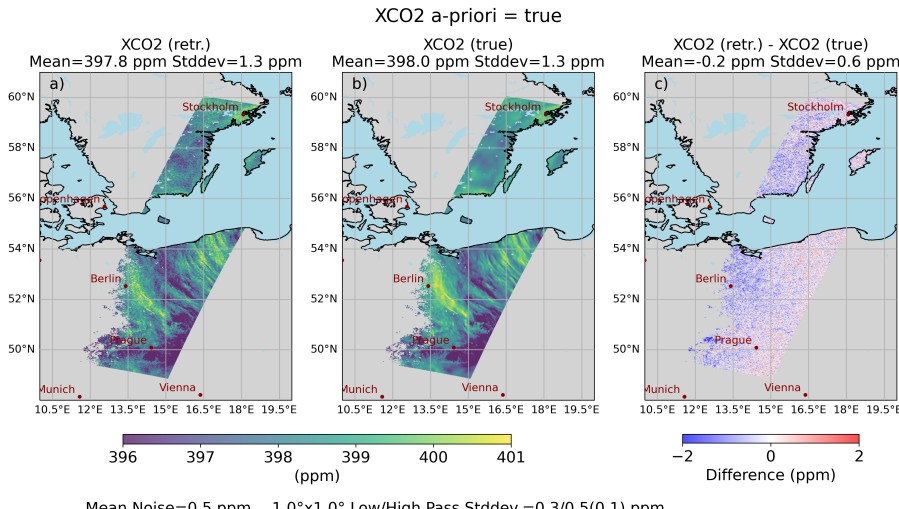

**Figure 5.** FOCAL-CO2M $XCO_2$ retrieval results for the 'Berlin scene' (only cloud-free data over land). a) Retrieved $XCO_2$. b) True $XCO_2$. c) Difference Retrieved - True $XCO_2$. The same post-processing filtering has been applied to all data shown in the plots. The number in brackets after the high-pass standard deviation gives an estimate for the high-pass standard deviation without noise.

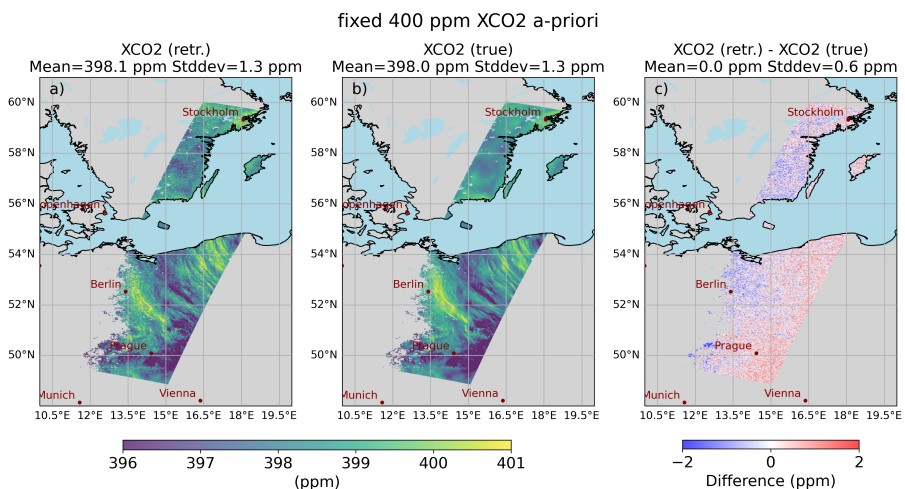

**Figure 6.** As Fig. 5, but for a fixed 400 ppm a-priori





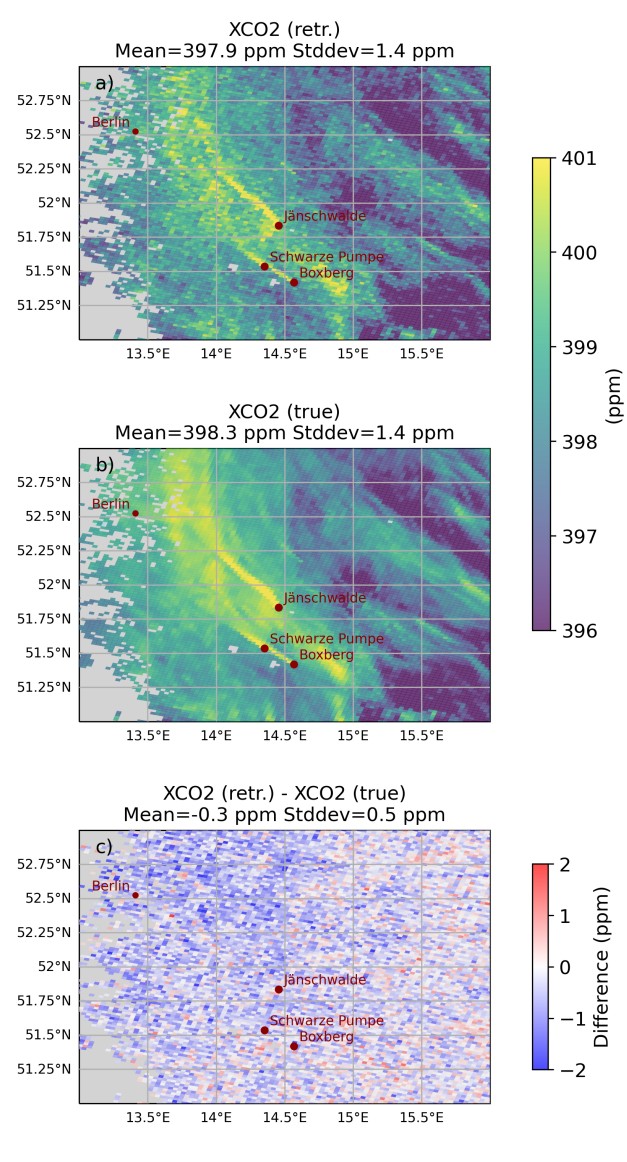

**Figure 7.** Zoom of Fig. 5.





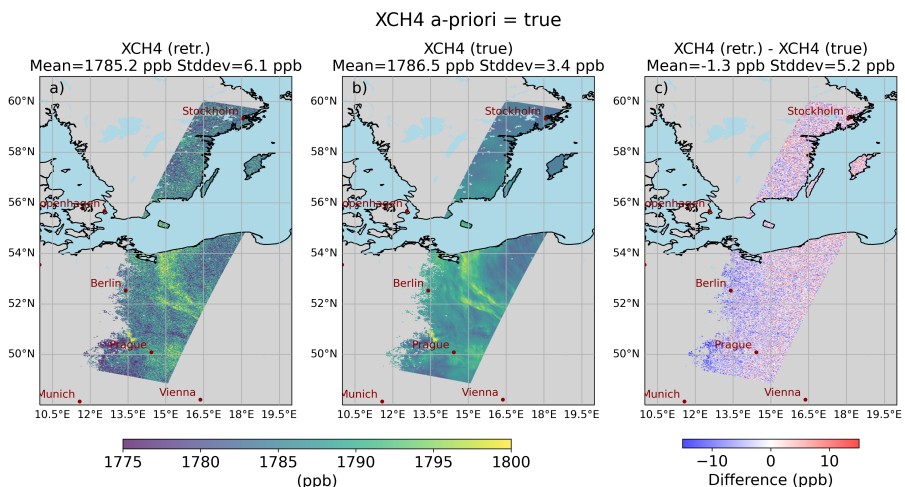

**Figure 8.** As Fig. 5, but for FOCAL-CO2M $XCH_4$.

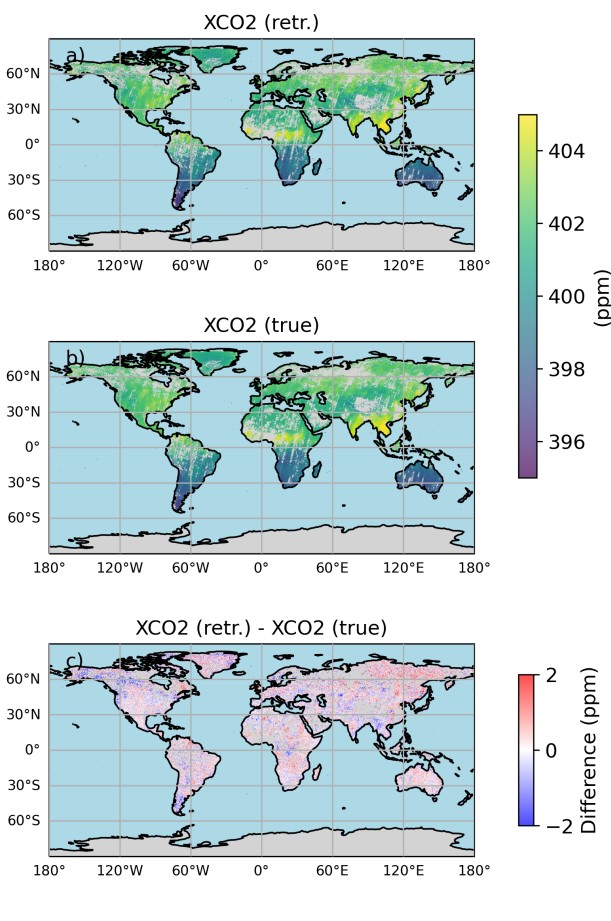

**Figure 9.** FOCAL-CO2M $XCO_2$ retrieval results for April 2015 (cloud-free subset data over land). a) Retrieved $XCO_2$. b) True $XCO_2$. c) Difference Retrieved - True $XCO_2$.





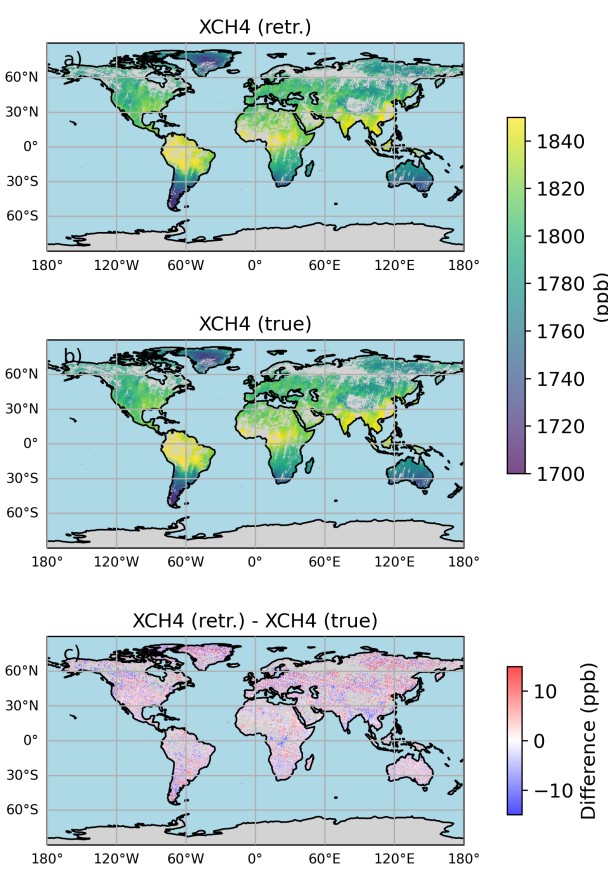

**Figure 10.** As Fig. 9, but for $XCH_4$.

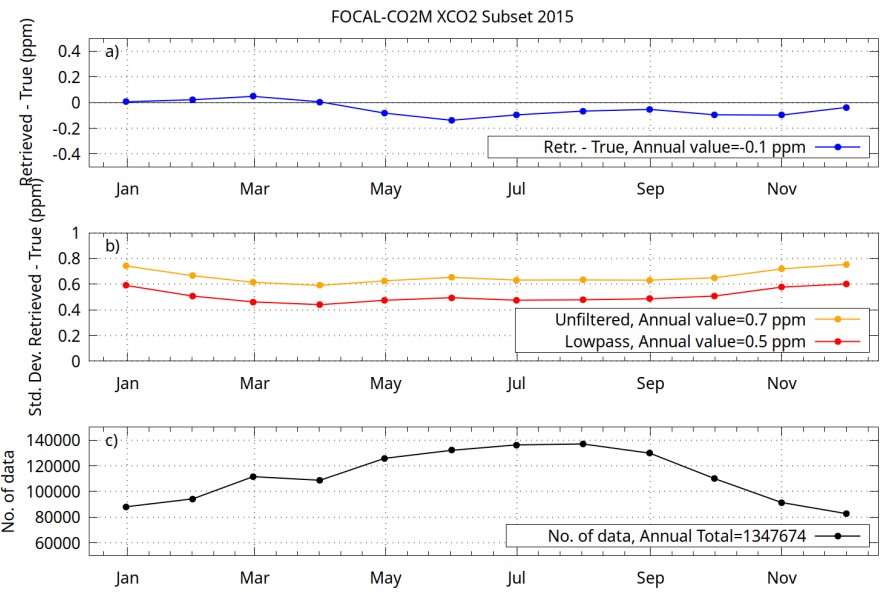

**Figure 11.** Monthly means and standard deviations of 2015 global subset data. a) Mean difference retrieved – true $XCO_2$. b) Standard deviation retrieved – true $XCO_2$ (orange: unfiltered, red: low-pass filtered). c) Number of data after post-processing. Annual values are given in the labels.





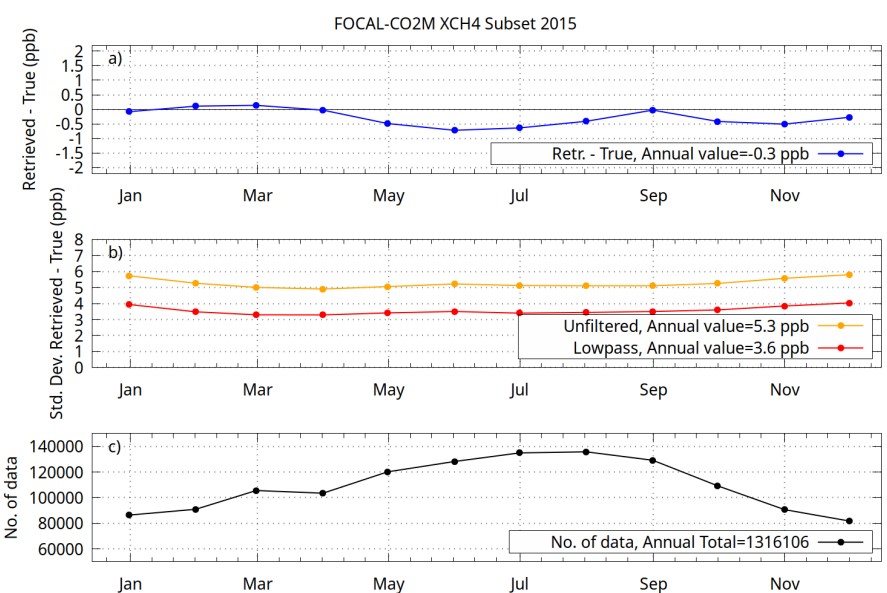

**Figure 12.** As Fig. 11, but for $XCH_4$.





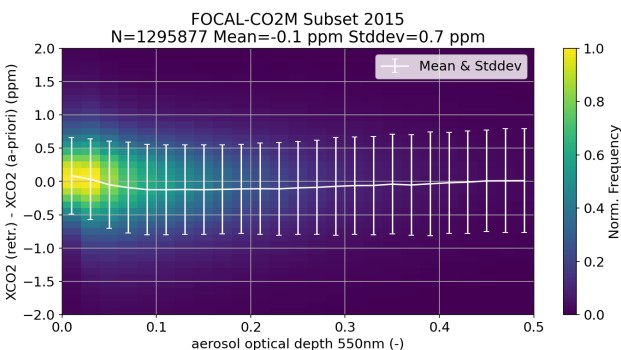

**Figure 13.** Difference between retrieved and true $XCO_2$ as function of AOD at $550\,\text{nm}$.





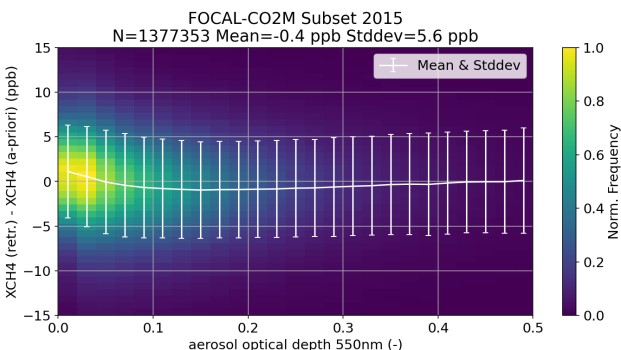

**Figure 14.** As Fig. 13, but for XCH$_4$.



**Table 1.** CO2M instruments and their characteristics.

| CO2I (Imaging Spectrometer) | | |
|---|---|---|
| Band | Spectral range | Spectral resolution |
| VIS | 405 – 490 nm | 0.6 nm |
| NIR | 747 – 773 nm | 0.12 nm |
| SWIR-1 | 1590 – 1675 nm | 0.3 nm |
| SWIR-2 | 1990 – 2095 nm | 0.35 nm |
| MAP (Multi-Angle Polarimeter) | | |
| Band | Central wavelength | Band width |
| VNIR-1 | 410 nm | 20 nm |
| VNIR-2 | 443 nm | 20 nm |
| VNIR-3 | 490 nm | 20 nm |
| VNIR-4 | 555 nm | 20 nm |
| VNIR-5 | 670 nm | 20 nm |
| VNIR-6 | 753 nm | 9 nm |
| VNIR-7 | 865 nm | 40 nm |
| CLIM (Cloud Imager) | | |
| Band | Band centre | Band width |
| CLIM-1 | 670 nm | 20 nm |
| CLIM-2 | 753 nm | 9 nm |
| CLIM-3 | 1370 nm | 15 nm |





**Table 2.** Parameters of instrument noise model. Unit of $A$ is $10^{-7} \, \mathrm{photons}^{-1} \, \mathrm{s} \, \mathrm{nm} \, \mathrm{cm}^2 \, \mathrm{sr}$.

| Parameter | NIR | SWIR-1 | SWIR-2 |
|---|---|---|---|
| A | 0.2 | 1.32 | 1.54 |
| B | 140 | 450 | 450 |



**Table 3.** Definition of FOCAL-CO2M spectral fit windows. Cross sections are from HITRAN2016 (Gordon et al., 2017, downloaded on 23 March 2021).

| No. | Name | Wavelength range (nm) | Considered gases |
|-----|------|----------------------|------------------|
| 1 | SIF | 747.0 – 759.0 | $O_2$, $H_2O$ |
| 2 | $O_2$ | 759.2 – 773.0 | $O_2$, $H_2O$ |
| 3 | Weak $CO_2$ | 1590.0 – 1670.0 | $CO_2$, $H_2O$, $CH_4$ |
| 4 | Strong $CO_2$ | 1990.0 – 2090.0 | $CO_2$, $H_2O$, $CH_4$ |



**Table 4.** State vector elements and related retrieval settings. A priori values are also used as first guess. "Fit windows" lists the spectral windows (see Table 3) from which the element is determined. "each" means that a corresponding element is fitted in each fit window. A priori values labelled as "PP" are taken from the provided meteorological data; "est." denotes that they have been estimated from the background signal.

| Element | Fit windows | A priori | A priori uncertainty | Comment |
|---------|-------------|----------|----------------------|---------|
| | | Gases / SIF | | |
| co2_lay | 3,4 | PP | 10.0 | $CO_2$ profile (5 layers), in ppm |
| ch4_lay | 3,4 | PP | 0.045 | $CH_4$ profile (5 layers), in ppm |
| h2o_lay | 3,4 | PP | 5.0 | $H_2O$ profile (5 layers), in ppm |
| sif_fac | 1 | 0. | 5. | SIF spectrum scaling factor |
| | | Scattering parameters | | |
| pre_sca | 1–4 | 0.2 | 1. | Layer height (rel. pressure, 0=surface, 1=infinity) |
| tau_sca_0 | 1–4 | 0.01 | 1. | Optical depth |
| ang_sca | 1–4 | 4.0 | 1. | Ångström coefficient |
| | | Polynomial coefficients (surface albedo) | | |
| poly0 | each | est. | 0.1 | estimated surface albedo |
| poly1 | each | 0.0 | 0.01 | |
| poly2 | each | 0.0 | 0.01 | |
| poly3 | each | 0.0 | 0.01 | |
| | | Spectral corrections | | |
| wav_shi | each | 0.0 | 0.1 | Wavelength shift |
| wav_squ | each | 0.0 | 0.001 | Wavelength squeeze |
| ils_squ | each | 1.0 | 0.1 | Slit function squeeze |