# Peer review of "Greenhouse Gas Retrievals for the CO2M mission using the FOCAL method: First Performance Estimates"

_Atmospheric Measurement Techniques, 2023_

## Referee Comment (RC1)

**Review of amt-2023-194**

The authors present performance estimates for their retrieval algorithm, FOCAL, applied to the upcoming CO2M mission. The performance estimates are based upon more detailed radiative transfer simulations using SCIATRAN, and are focused on the core retrieval algorithm biases rather than instrumental, spectroscopic, or meteorological. The authors estimate that FOCAL in it's current form meets (just!) the extremely stringent XCO2 and XCH4 requirements that have been placed on CO2M over land.

Overall the paper is well written, provides useful information, and is well suited for AMT. However I have concerns about how representative the bias correction method is when being trained on the exact truth. More justification and information on the bias correction should be included, and conclusions updated accordingly. Specific comments follow below.

**General Comments**

I recognize the need for bias correction, however I think what is done here may go beyond what is typical and requires at minimum more justification. The authors train their bias correction scheme against actual true values, which will not be known for CO2M. No results of the bias correction are shown, except it is clear that it likely has an impact on their final uncertainty quantification for large-scale fluxes based upon Figure 11 showing the best performance for the one month of data (April) that the bias correction is trained on. I would like to see more detail on what effect the bias correction is having, as well as what parameters it has deemed important. After all since this is entirely simulated data the results of the bias correction should be fully explainable. I realize the authors state that their results "do not consider any systematic errors in meteorology" but what this means in terms of the bias correction is missing.

In some places I find the conclusions of the authors a little optimistic. For example the last sentence of the manuscript is "However, the current results give good confidence that the FOCAL-CO2M retrieval is able to fulfil the product quality requirements of the CO2M mission." But my interpretation of the results is if you neglect things not taken into account in the simulation (polarization, full sphericity, 2d/3d effects, among others), assume you have perfect spectroscopy and input ancillary data, assume you have a perfect instrument, and perform bias correction using perfect truth values, then you barely meet the requirements. This is not a knock on the retrieval method, the CO2M requirements are very stringent, but it is hard for me to have a takeaway other than that it is unlikely FOCAL will meet the requirements when applied to real CO2M data in its current form.

**Specific Comments**

**p.3 l.57:**
I realize that a full description of the three algorithms is beyond the scope of this paper, and that they can be found in the corresponding citations, but a brief statement here about their differences would be helpful for the reader.

**p.3 l.78:** "The SCIATRAN calculations are more complex than the FOCAL forward model. For example, they consider surface BRDF (bidirectional reflectance distribution function) effects, different aerosol types and distributions as well as clouds."
This needs some further elaboration. While SCIATRAN itself can be much more complicated than the FOCAL forward model, it may not be configured that way. I see later on that polarization is neglected in the SCIATRAN calculations for example. I assume other features of SCIATRAN such as line of sight sphericity are not included even though they are supported as well.

**p.5 l.143:** "we filter out all cloudy data"
Surely not every scene with any clouds are filtered out or else there would be no purpose in including clouds

in the SCIATRAN runs?

**p.6 l.171:** "based on a set of training and test data"
Most algorithms now recommend splitting data into three groups: Train, test, and validation. The reasoning being that you can overfit to your "test" data by tweaking your model parameters, or even what model you are using.

**p.6 l.183:** "bias correction does not consider any additional errors resulting from systematic differences between the estimated meteorological conditions and the actual atmosphere"
Could this be included in some way? I see from previous publications that for real data truth is taken from a generated database. Is there no error estimate for this database that could be artificially included here?

**p.7 l.204:** "...cloud-free data ..."
As before, do you mean cloud VOD is identically 0? Or are some thin clouds included?

**Section 4.1:** For the plume specifically we are probably getting down to the scale where 2d/3d effects are important. I don't expect the authors to quantify these effects since I know SCIATRAN is not capable of doing so and it is a large amount of work, but I was surprised to see no mention of the effect

**Section 4.3:** Does the bias correction use the FOCAL retrieved optical depth in a significant way? I assume the idea would be to use the MAP aerosol parameters in the bias correction instead. Could you not test this directly by using the true aerosol OD + estimated errors of the MAP retrieval in the bias correction?

**Technical Corrections**

**p.2 l.45:** "welling" → "upwelling"

**p.2 l.53:** I would recommend using "better" here instead of "higher", since "higher spatial resolution" can be ambiguous.

**p.9 l.255:** "is not much sensitive"
Reword

**p.9 l.257:** "an" → "any"

---

## Referee Comment (RC2)

**Review of the manuscript**

*"Greenhouse Gas Retrievals for the CO2M mission using the FOCAL method: First Performance Estimates"*

**Comprehension**

The manuscript presents an assessment of the FOCAL-CO2M retrieval algorithm's capabilities in quantifying CO2 and CH4 enhancements from space. The study communicates its limitations, is concise and articulated with good clarity throughout most parts. However, there are some aspects that should be improved.

**General comments**

The SCIATRAN setup does not fully represent the complex and varied conditions of the actual atmosphere. While useful for controlled testing and isolating error analysis it raises questions about the algorithm's performance in real-world scenarios, especially under unpredictable atmospheric conditions. I am concerned that the budget, especially in terms of the mean noise error, could be depleted if we incorporate additional error sources.

I'd like to see a more in-depth exploration of the limitations. To examine meteorology's influence, one could alter variables such as humidity and temperature profiles in SCIATRAN, aligning them with their respective uncertainties.

Are there alternative reference spectra for CO2M that employ a more sophisticated instrument model? Assuming a perfect instrument model presents a limitation.

It might be beneficial to create a distinct section for the post-processing phase, as it involves more than just basic threshold filtering, unlike the pre-processing step.

Does SCIATRAN's treatment of aerosols or the aerosol input include assumptions or simplifications that are advantageous to the FOCAL setup? Additional details would aid in interpreting the results, especially regarding Figures 13 and 14.

What is the consequence of an imperfect cloud screening during the pre-processing phase? In reality, there may be subpixel residual clouds (smaller than 400 x 400 m2) that go undetected.

I can't find a discussion how a priori estimates affect the error margins in the FOCAL retrieval. Are the priors used in the FOCAL fit, other than the target quantities, equivalent to their true values?

Once real data becomes available in 2026, what will the post-processing strategy entail? Will parameters be computed on a monthly, quarterly, or yearly basis? Could you provide insights into any existing plans or considerations on this matter?

What is the rationale behind selecting a 5-layer profile fit? What kind of analysis lead to this choice. Could you provide information on the errors associated with each of these layers?

To what extent is the 400 ppm a priori profile scaled in each atmospheric layer, and does the lowest layer contain the majority of the enhancement? Additionally, what advantages does fitting five layers offer, particularly in the study of anthropogenic emissions, when it's known that most emissions are concentrated in the lower levels?

A more fundamental error analysis such as information on the reduction in uncertainty from priori to the a posteriori state would be appreciated.

Is the term "full physics" (FP) appropriate when the forward model in FOCAL is notably simpler than models like SCIATRAN and relies on approximations and parameterizations for various physical aspects? Additionally, considering that bias correction in the post-processing step is primarily necessary due to the limitations of the forward model, the use of the term "full physics" may be open to question. Nevertheless, I understand that it's a recognized term within the scientific community.

**Specific comments**

5: XCO2 and XCH4, but no XNO2?

15: Any reason for choosing 2015?

15: Maybe "caused" instead of "due"?

19-20: Does this imply that the mission requirements could potentially be met without the MAP instrument?

26: Is "However, …" applicable here? Consider rephrasing.

41-44: In principle, would a single satellite be sufficient to fulfill the mission? Is the MVS considered operational with just one CO2M satellite instrument in orbit?

125: Is the thickness of the single scattering layer dependent on the vertical atmospheric grid utilized in the forward model? Could you please specify the thickness of the layer in the context of this study?

143-144: Does this require an updated FOCAL version which includes MAP instrument data?

145: Why is a single Signal-to-Noise Ratio (SNR) threshold established when the requirement on line 64 specifies different reference albedos for various channels? Additionally, is the average SNR within each band used, or is the SNR specified at a particular channel position taken into account?

166: Does this paragraph refer to the second filtering step?

170: This should be mentioned earlier, maybe already around line 164 or 166.

175-179: Consider to improve clarity.

195: Consider providing additional details on how the forward model error differs when applied to real data compared to theoretical or simulated data.

241-243: Consider to rephrase, in particularly "It should be noted …" as is not very clear to me.

242-243: The standard deviation after application of filters is 0.2 and 0.5 ppm. Confirm that the latter represents the mean noise error? Also confirm that the smaller mean difference value suggests that the high-pass standard deviation is dominated by (random) noise?

263: Please confirm whether the 0.5 ppm standard deviation, observed after applying filters, represents the mean noise error. Also, does the smaller mean difference value indicate that the high-pass standard deviation is predominantly influenced by random noise?

Fig. 8: Do you have an estimate of the extent to which an inaccurate prior (a priori that doesn't represent the truth) might impact the fitting of CH4 on the various levels?

278-280: Is there a specific reason why a recalculation of the post-processing database for the entire year 2015 is not being conducted? Is it due to cost constraints, technical limitations, or other factors?

Fig. 9: Any idea, what causes the positive or negative biases in some areas?

---

## Author Comment (AC1)

Greenhouse Gas Retrievals for the CO2M mission using the FOCAL method: First Performance Estimates
by S. Noël et al.

MS No.: amt-2023-194

**Reply to Referee 1**

We thank the referee for the review and the comments. They will be considered in the revised version of the paper. In the following, the original reviewer comments are given in *italics*, our answer in normal font.

*The authors present performance estimates for their retrieval algorithm, FOCAL, applied to the upcoming CO2M mission. The performance estimates are based upon more detailed radiative transfer simulations using SCIATRAN, and are focused on the core retrieval algorithm biases rather than instrumental, spectroscopic, or meteorological. The authors estimate that FOCAL in it's current form meets (just!) the extremely stringent XCO2 and XCH4 requirements that have been placed on CO2M over land. Overall the paper is well written, provides useful information, and is well suited for AMT. However I have concerns about how representative the bias correction method is when being trained on the exact truth. More justification and information on the bias correction should be included, and conclusions updated accordingly. Specific comments follow below.*

We will add more information about the bias correction and update the conclusions, see also following answers.

**General Comments**

1. *I recognize the need for bias correction, however I think what is done here may go beyond what is typical and requires at minimum more justification. The authors train their bias correction scheme against actual true values, which will not be known for CO2M. No results of the bias correction are shown, except it is clear that it likely has an impact on their final uncertainty quantification for large-scale fluxes based upon Figure 11 showing the best performance for the one month of data (April) that the bias correction is trained on. I would like to see more detail on what effect the bias correction is having, as well as what parameters it has deemed important. After all since this is entirely simulated data the results of the bias correction should be fully explainable. I realize the authors state that their results "do not consider any systematic errors in meteorology" but what this means in terms of the bias correction is missing.*

   The bias correction is indeed important to meet the requirements (this is essentially common standard for greenhouse gas retrieval algorithms).

   We use the a-priori model data as truth for the bias correction, because this is also planned to be done for real measurements data. Since the a-priori data are also the truth for the simulated data we agree with the referee that we do not consider here additional errors due to the limited knowledge of the truth. However, these errors strongly depend on the quality of the a-priori data which can only be determined by e.g. validation based on real data. Therefore we decided not to include this additional uncertainty. We will emphasise this in the updated manuscript.

   We will also add more information about the post-processing (filtering and bias correction), e.g. on which parameters it is based, and we will add a map of the bias correction.

   Note that the uncertainty quantification for large-scale fluxes shown in Figure 11 is not only based on the April 2015 data. The figure shows the results for all months of the year, i.e. also for data not used for training. We therefore consider our conclusions to be valid (under the mentioned assumptions).

The impact of systematic errors in meteorology on the bias correction cannot be reliably quantified, because this depends on the quality of the input meteorological data. In principle, errors in meteorology affect the data quality, but currently we assume that these data are on average correct and therefore their uncertainties for specific conditions should not have a major impact on the bias correction, especially because meteorological data are (intentionally) not used as parameters / features.

In this context, we would like to emphasise that FOCAL has been applied to real satellite data using similar bias correction as used and described in this paper, where we write in section 3.1:

"Applications to OCO-2, GOSAT and GOSAT-2 have shown that FOCAL is fast and produces accurate results. For example, the spatio-temporal bias of the FOCAL XCO2 product derived from TCCON comparisons is (after bias correction) in the order of 0.6 ppm for OCO-2 (Reuter and Hilker, 2022), and 0.6 (1.1) ppm for GOSAT (GOSAT-2) (Noël et al., 2022)."

This gives confidence that meaningful results for CO2M are presented in this publication although all results are based only on simulations and despite the fact that not all error sources are addressed in this publication.

2. *In some places I find the conclusions of the authors a little optimistic. For example the last sentence of the manuscript is "However, the current results give good confidence that the FOCAL-CO2M retrieval is able to fulfil the product quality requirements of the CO2M mission." But my interpretation of the results is if you neglect things not taken into account in the simulation (polarization, full sphericity, 2d/3d effects, among others), assume you have perfect spectroscopy and input ancillary data, assume you have a perfect instrument, and perform bias correction using perfect truth values, then you barely meet the requirements. This is not a knock on the retrieval method, the CO2M requirements are very stringent, but it is hard for me to have a takeaway other than that it is unlikely FOCAL will meet the requirements when applied to real CO2M data in its current form.*

It is true that our simulations of the spectra do not consider all possible physical processes. However, it is not possible to include all of these in the simulations, because even if the radiative transfer model would be able to consider this the required input data on e.g. 2d/3d cloud or aerosol distribution is not available. We will mention this in the updated paper.

Regarding polarisation:
Some test data calculated by RAL (see reference in the paper) are available with/without considering polarisation in the radiative transfer. We have tested the impact of polarisation on our retrieval using these data, which is for XCO2 in the order of 0.1 ppm before bias correction. Therefore we think that neglecting polarisation in the SCIATRAN runs can be justified.

Regarding sphericity:
Both SCIATRAN and FOCAL could consider (at least pseudo) sphericity; however, in case of CO2M (normally nadir looking with a swath of about 200 km) spherical geometry has no major impact for solar zenith angles less than 75 deg and is therefore neglected.

The main intention of the manuscript is to show that the FOCAL-CO2M retrieval method is in principle able to fulfil the requirements. We cannot finally state now that we will fulfil the requirements for real data. Especially for large scale fluxes we are indeed on the edge, but this is not the primary objective of CO2M. We will try to clarify this more in the updated paper.

Nevertheless, we think that the current results – under the mentioned assumptions / limitations – show that we are on a good way (which is why we chose the "good confidence" formulation). For the revised version of the paper we removed "good". The modified sentence now reads as follows:

"However, the current results give confidence that the FOCAL-CO2M retrieval algorithm will be able to generate products meeting the product quality requirements of the CO2M mission."

**Specific Comments**

1. *p.3 l.57: I realize that a full description of the three algorithms is beyond the scope of this paper,*

*and that they can be found in the corresponding citations, but a brief statement here about their differences would be helpful for the reader.*

We will add some more information about the algorithms.

2. *p.3 l.78: "The SCIATRAN calculations are more complex than the FOCAL forward model. For example, they consider surface BRDF (bidirectional reflectance distribution function) effects, different aerosol types and distributions as well as clouds." This needs some further elaboration. While SCIA-TRAN itself can be much more complicated than the FOCAL forward model, it may not be configured that way. I see later on that polarization is neglected in the SCIATRAN calculations for example. I assume other features of SCIATRAN such as line of sight sphericity are not included even though they are supported as well.*

We will include some more information about the SCIATRAN calculations.

See also our response above regarding polarisation and sphericity.

3. *p.5 l.143: "we filter out all cloudy data" Surely not every scene with any clouds are filtered out or else there would be no purpose in including clouds in the SCIATRAN runs?*

We consider clouds in the SCIATRAN calculations, but for the performance tests used in this study we only consider completely cloud-free soundings. We will clarify this in the text.

4. *p.6 l.171: "based on a set of training and test data" Most algorithms now recommend splitting data into three groups: Train, test, and validation. The reasoning being that you can overfit to your "test" data by tweaking your model parameters, or even what model you are using.*

The bias correction is determined only from the training data. The test data are used to check if the performance on this data set is similar to the performance for the training set. Both training and test data are taken from the April 2015 results. In addition, we apply later on the bias correction to the complete 2015 data (with good results, as shown in the paper), so the additional 11 months can be considered as a validation data set.

5. *p.6 l.183: "bias correction does not consider any additional errors resulting from systematic differences between the estimated meteorological conditions and the actual atmosphere" Could this be included in some way? I see from previous publications that for real data truth is taken from a generated database. Is there no error estimate for this database that could be artificially included here?*

As described above, it is not possible to include systematic differences between the estimated meteorological conditions and the actual atmosphere in the bias correction, because this would require a knowledge of specific systematic errors of the meteorological data, which are usually unknown (otherwise there would be a correction for them). If – as we assume – the meteorological data are on average correct, statistical (random-like) uncertainties in the data would have no impact on the bias correction.

Previous applications of the FOCAL algorithm to e.g. GOSAT and GOSAT-2 use indeed as "true" $XCO_2$ and $XCH_4$ for the bias correction a database (e.g. derived from the SLIM climatology). We could also use the SLIM climatology for CO2M, but we assume that the actual CAMS model data which will be provided during the mission are a better estimate for the true $XCO_2$ and $XCH_4$. This is why we use the CAMS data as truth in this study.

The bias correction will always try to reproduce the values given as truth. Random errors in the assumed truth are less relevant in this context, but any systematic error in the assumed true $XCO_2$ and $XCH_4$ will result in a corresponding error of the retrieved products. The latter error can only be quantified by comparisons with independent data (e.g. ground based measurements). There are error estimates for SLIM but these are not applicable to the operational CAMS data and therefore could only be used as a rough indication for possible additional errors due to the uncertainties of the "truth".

6. *p.7 l.204: ". . . cloud-free data . . . " As before, do you mean cloud VOD is identically 0? Or are some thin clouds included?*

   As mentioned above, for the performance verification we use only completely cloud-free data.

7. *Section 4.1: For the plume specifically we are probably getting down to the scale where 2d/3d effects are important. I don't expect the authors to quantify these effects since I know SCIATRAN is not capable of doing so and it is a large amount of work, but I was surprised to see no mention of the effect*

   We will mention that 2d/3d effects are not explicitly considered in the radiative transfer simulations.

8. *Section 4.3: Does the bias correction use the FOCAL retrieved optical depth in a significant way? I assume the idea would be to use the MAP aerosol parameters in the bias correction instead. Could you not test this directly by using the true aerosol OD + estimated errors of the MAP retrieval in the bias correction?*

   The derived parameters of the scattering layer (incl. optical depth) are used in the bias correction (will be shown in updated paper, see above). These parameters correlate quite well with aerosol. We did some tests using the true AOD as possible additional parameter for the bias correction, but this did not significantly change the results.

   There is not much information available about the quality of the retrieved MAP aerosol parameters, so we were not able to use this, especially since we would need this MAP L2 output consistent with our input data.

   However, the inclusion of MAP data for e.g. bias correction is foreseen in the algorithm, but this can only be tested with real data.

**Technical Corrections**

1. *p.2 l.45: "welling" → "upwelling"*

   Corrected.

2. *p.2 l.53: I would recommend using "better" here instead of "higher", since "higher spatial resolution" can be ambiguous.*

   Changed.

3. *p.9 l.255: "is not much sensitive" Reword*

   Done, new text:
   "To show that the sensitivity of the retrieval to the choice of the a-priori is low, ..."

4. *p.9 l.257: "an" → "any"*

   Changed.

---

## Author Comment (AC2)

**Greenhouse Gas Retrievals for the CO2M mission using the FOCAL method: First Performance Estimates**
by S. Noël et al.

MS No.: amt-2023-194

**Reply to Referee 2 (Philipp Hochstaffl)**

We thank the referee for the review and the comments. They will be considered in the revised version of the paper. In the following, the original reviewer comments are given in *italics*, our answer in normal font.

**Comprehension**

*The manuscript presents an assessment of the FOCAL-CO2M retrieval algorithm's capabilities in quantifying CO2 and CH4 enhancements from space. The study communicates its limitations, is concise and articulated with good clarity throughout most parts. However, there are some aspects that should be improved.*

**General comments**

1. *The SCIATRAN setup does not fully represent the complex and varied conditions of the actual atmosphere. While useful for controlled testing and isolating error analysis it raises questions about the algorithm's performance in real-world scenarios, especially under unpredictable atmospheric conditions. I am concerned that the budget, especially in terms of the mean noise error, could be depleted if we incorporate additional error sources. I'd like to see a more in-depth exploration of the limitations. To examine meteorology's influence, one could alter variables such as humidity and temperature profiles in SCIATRAN, aligning them with their respective uncertainties.*

   We agree with the referee, that the SCIATRAN simulations (as any simulations) are not fully representative for real-world conditions. Indeed, we expect to have larger systematic errors for real measurements, because many error sources have not been considered. This is why we list all these limitations and explicitly do not state that we will fulfil the requirements for real measurements, but we think we are on a good way, see e.g. the last sentence of the conclusions:

   "the current results give good confidence that the FOCAL-CO2M retrieval is able to fulfil the product quality requirements of the CO2M mission."

   Regarding meteorology, we assume in the retrieval (also for real data) that the provided a-priori data are on average correct, which is a reasonable assumption for CAMS data. Of course, for single soundings meteorology will not be perfect, which will then result in an additional bias for the retrieved data products. This will then increase the scatter of the data and therefore affect also the systematic error estimate, which is derived from the standard deviation of the bias. Because these are noise-like errors, we expect no large effect on the global/continental scale (low-pass filtered) data. For the local scale (high-pass filtered data), this has an impact, but these data are dominated by (non-systematic) instrumental noise, which leaves some margin for additional errors. Therefore we think that uncertainties in the meteorological data would not much affect the statistical results for the systematic errors.

   Note that – since we are dealing with statistical quantities here – it is in general not possible to quantify the effect of uncertainties in the meteorology on the systematic errors by a limited set of tests with varying variables. We would have to re-produce at least a month of spectral data (probably more,

depending on how many variables we vary), process them, update the post-processing and apply it. Then, we would still not be sure how representative our simulations are for real measurements.

Therefore, we think it is better to assess these issues with real measurements.

As written in the title of the paper, we only present a first performance assessment here, which just shows that our algorithm is capable to fulfil the requirements for a specific set of test data.

2. *Are there alternative reference spectra for CO2M that employ a more sophisticated instrument model? Assuming a perfect instrument model presents a limitation.*

   We think that the used instrument model is appropriate for the application addressed in this manuscript, where the focus is on retrieval algorithm related errors. Note that instrument related errors have been investigated in detail in the context of several ESA studies (e.g., Buchwitz et al., 2020), resulting in detailed instrument requirements (ESA, 2020). (All references are given in the paper.)

3. *It might be beneficial to create a distinct section for the post-processing phase, as it involves more than just basic threshold filtering, unlike the pre-processing step.*

   We will add more information about post-processing (especially bias correction) in the revised version of the paper.

4. *Does SCIATRAN's treatment of aerosols or the aerosol input include assumptions or simplifications that are advantageous to the FOCAL setup? Additional details would aid in interpreting the results, especially regarding Figures 13 and 14.*

   In general, aerosols are described in SCIATRAN by microphysical parameters and specific scattering functions. In FOCAL, only a single effective scattering layer is considered which has to account for all kinds of scattering including Rayleigh, aerosols, and scattering of residual clouds if present. We also do not use any external a-priori information on aerosols in FOCAL. So, the treatment of aerosols is completely different.

   We will add some more information on aerosol treatment by SCIATRAN in the revised paper.

5. *What is the consequence of an imperfect cloud screening during the pre-processing phase? In reality, there may be subpixel residual clouds (smaller than 400 x 400 m2) that go undetected.*

   For the simulated data we currently only have fully cloudy or cloud-free scenes, no partially cloudy scenes. Therefore we cannot give quantitative information about the impact of subpixel residual clouds. However, we agree that we would need a quite strict cloud filtering for real data and still there could be remnant clouds. From experience with other satellite missions we know that this problem can be handled, e.g. by an additional cloud filtering method. Real data are needed to make this decision.

   Since the requirements on systematic errors are applicable to cloud-free scenarios, undetected clouds do not affect the fulfilment of the requirements.

6. *I can't find a discussion how a priori estimates affect the error margins in the FOCAL retrieval. Are the priors used in the FOCAL fit, other than the target quantities, equivalent to their true values?*

   For CO2, CH4 and H2O we use the (true) CAMS values as a-priori. The a-priori for the surface albedo is estimated from the measured reflectances. All other a-priori values are fixed. This information is given in Table 4 (state vector elements).

   As shown in the paper, the retrieval results for a fixed 400 ppm CO2 a-priori are very similar to the ones using the true values as a-priori. We also assume quite large a-priori errors (see the discussion on uncertainty reduction later). This indicates that the choice of the a-priori does not affect the estimation of systematic errors in a significant way.

7. *Once real data becomes available in 2026, what will the post-processing strategy entail? Will parameters be computed on a monthly, quarterly, or yearly basis? Could you provide insights into any existing plans or considerations on this matter?*

To our knowledge, this strategy is not fixed yet. As written in the paper, there will be an update of the post-processing data base during the commissioning phase. The decision if later updates are required has to be taken based on validation and monitoring results.

Note that for our GOSAT data processing with FOCAL (see references in paper) a single post-processing data base determined from one year of data was successfully applied to the whole time series 2009–2021.

8. *What is the rationale behind selecting a 5-layer profile fit? What kind of analysis lead to this choice. Could you provide information on the errors associated with each of these layers?*

Nadir measurements have only a limited vertical information, there are typically only 2-3 independent information points for CO2 and CH4 profiles. There is therefore no use in a large number of retrieval layers. Our choice of 5 layers (each containing the same number of particles) is based on the experience with previous missions (OCO-2, GOSAT, GOSAT-2) and simulations (Reuter et al. FOCAL papers, references in the paper). Note that pressure and temperature profiles used in the FOCAL forward model have 20 layers.

We get uncertainties for each layer, these are then combined in the total XCO2 and XCH4 uncertainty. This is provided together with the XCO2 and XCH4 and corresponding averaging kernel matrices in the L2 data product. The profiles themselves are not part of the operational data product, and there are no requirements on these, therefore there is no need to discuss them in the paper.

9. *To what extent is the 400 ppm a priori profile scaled in each atmospheric layer, and does the lowest layer contain the majority of the enhancement? Additionally, what advantages does fitting five layers offer, particularly in the study of anthropogenic emissions, when it's known that most emissions are concentrated in the lower levels?*

Although the data product is only the total column, fitting a (coarse) profile has proven to give more accurate results than simply scaling an a-priori profile. This is especially useful for anthropogenic emissions, where the enhancement is at the lower altitudes. In case of the plumes, the retrieved enhancement for the 400 ppm a-priori is indeed only visible in the two lowest layers.

10. *A more fundamental error analysis such as information on the reduction in uncertainty from priori to the a posteriori state would be appreciated.*

The total a-posteriori error of XCO2 (XCH4) is about 0.6 ppm (5.8 ppb), i.e. very similar to the corresponding noise errors. This compares to an assumed a-priori uncertainty (see table 4) of 5 ppm (corrected number) for CO2 and 45 ppb for CH4. This means the CO2 uncertainty is reduced by the retrieval by a about a factor of 8.

We will add this information in the paper.

11. *Is the term "full physics" (FP) appropriate when the forward model in FOCAL is notably simpler than models like SCIATRAN and relies on approximations and parameterizations for various physical aspects? Additionally, considering that bias correction in the post-processing step is primarily necessary due to the limitations of the forward model, the use of the term "full physics" may be open to question. Nevertheless, I understand that it's a recognized term within the scientific community.*

The term "full physics" is indeed not well defined, but it is commonly used. As written in the paper, we mainly use it to distinguish between the directly retrieved products and the proxy products derived from ratios of retrieved quantities and model data.

To our understanding, a FP model describes the propagation of light through the atmosphere while considering physical processes like absorption and scattering. Each model uses different assumptions and limitations or parameterisations, those for FOCAL are just different to those for e.g. SCIATRAN. Having a post-processing to filter/correct the retrieved values does not make a method "non physical". In fact, all greenhouse gas retrievals require a bias correction to meet the stringent requirements.

**Specific comments**

1. *l. 5: XCO2 and XCH4, but no XNO2?*

   Yes, NO2 is provided as total column, not column mixing ratio.

2. *l 15: Any reason for choosing 2015?*

   2015 was chosen because EUMETSAT provided the observational geometry data for this year.

3. *15: Maybe "caused" instead of "due"?*

   Will be changed.

4. *19-20: Does this imply that the mission requirements could potentially be met without the MAP instrument?*

   The current study shows that FOCAL-CO2M may (based on the specific set of simulated input test data) meet the requirements without MAP. However, due to the mentioned limitations, it is not sure if this will be the case for real data. In any case, we will investigate if the inclusion of MAP data (once available) will improve out results.

5. *26: Is "However, ..." applicable here? Consider rephrasing.*

   We will slightly rephrase this sentence.

6. *41-44: In principle, would a single satellite be sufficient to fulfill the mission? Is the MVS considered operational with just one CO2M satellite instrument in orbit?*

   In principle, monitoring of CO2 and CH4 is possible with a single satellite, but multiple satellites are beneficial to increase spatial and temporal coverage.

7. *125: Is the thickness of the single scattering layer dependent on the vertical atmospheric grid utilized in the forward model? Could you please specify the thickness of the layer in the context of this study?*

   The optical thickness (optical depth) of the scattering layer is derived during the fit (together with the layer height and the Ångstöm coefficient). All these quantities are not related to the layering of the atmosphere. In terms of geometrical thickness, the scattering layer in FOCAL is actually assumed to be infinitely thin.

   We will remove "thin" in this sentence to avoid a misleading interpretation.

8. *143-144: Does this require an updated FOCAL version which includes MAP instrument data?*

   Data over ocean / glint can be processed with the same FOCAL version. However, the post-processing data bases need to be adapted, this work is currently ongoing.

9. *145: Why is a single Signal-to-Noise Ratio (SNR) threshold established when the requirement on line 64 specifies different reference albedos for various channels? Additionally, is the average SNR within each band used, or is the SNR specified at a particular channel position taken into account?*

   The SNR limit is only used to (roughly) filter out data with low signal before the inversion. We use for this the signal at spectral regions outside of strong absorption (the chosen wavelengths given in the paper).

   This SNR filter is not related to the requirement on random errors, which is specified and only valid for a certain scenario.

10. *166: Does this paragraph refer to the second filtering step?*

    Yes, we will clarify this.

11. *170: This should be mentioned earlier, maybe already around line 164 or 166.*

    This paragraph describes the bias correction (which we already mention in line 156), the paragraphs before are related to filtering. Therefore we think this paragraph should stay at this position.

12. *175-179: Consider to improve clarity.*

    We will try to clarify this with references to the methods described above.

13. *195: Consider providing additional details on how the forward model error differs when applied to real data compared to theoretical or simulated data.*

    The forward model error is determined from the relation between the specified noise on the spectra and the fit residuals, i.e. it depends on the input data. We do not have any real data yet, so we do not know how this will change.

14. *241-243: Consider to rephrase, in particularly "It should be noted ..." as is not very clear to me.*

    We will try to clarify this in the revised version.

15. *242-243: The standard deviation after application of filters is 0.2 and 0.5 ppm. Confirm that the latter represents the mean noise error? Also confirm that the smaller mean difference value suggests that the high-pass standard deviation is dominated by (random) noise?*

    The latter (high-pass filtered) value contains systematic errors (those which we want to verify) as well as random (noise) errors. The noise error (given also in the plots) is mainly determined by the instrument noise and of similar magnitude as the high-pass filtered value. This indicates that the high-pass filtered value is dominated by noise and thus the systematic error is much lower.

    This is not related to the mean difference, which is not relevant for the standard deviation.

16. *263: Please confirm whether the 0.5 ppm standard deviation, observed after applying filters, represents the mean noise error. Also, does the smaller mean difference value indicate that the high-pass standard deviation is predominantly influenced by random noise?*

    See answer to previous question.

17. *Fig. 8: Do you have an estimate of the extent to which an inaccurate prior (a priori that doesn't represent the truth) might impact the fitting of CH4 on the various levels?*

    We have not explicitly checked the impact of e.g. a constant CH4 a-priori on the retrieval results, but we expect that this impact is as for CO2 rather small.

18. *278-280: Is there a specific reason why a recalculation of the post-processing database for the entire year 2015 is not being conducted? Is it due to cost constraints, technical limitations, or other factors?*

    We could have determined the post-processing data base using the measurements of the entire year as input. In fact, this would improve the results, because also seasonal variations could be taken into account. We decided to limit the input data for the post-processing to one month for the following reasons:

    (a) We wanted to be as close to "real" conditions as possible. During the commissioning phase we need to re-determine the post-processing data base, and there will be most likely only a limited amount of data available at that time.

    (b) With the current setup it is possible to show that the post-processing is working also for data / time periods which were not used during generation of the data base.

    We will add this information in the revised version.

19. *Fig. 9: Any idea, what causes the positive or negative biases in some areas?*

    This is not clear. Zooming into the figure you can see that there are usually positive and negative deviations at the same place. This confirms that a significant contribution to the variation is due to noise. Some areas seem to have overall a more positive or negative bias. The reasons for this are currently unclear. This could be related to surface properties or maybe aerosols, but there seems to be no systematic in there, otherwise the bias correction would have probably handled this.

---

## Referee Report (RR1)

**Review of the REVISED manuscript**

**"Greenhouse Gas Retrievals for the CO2M mission using the FOCAL method: First Performance Estimates"**

**General comments**

The refined version improves the FOCAL-CO2M retrieval algorithm's assessment and should be subject to publication. The authors have enhanced the manuscript by addressing several key issues highlighted in the review. I appreciate and thorough responses in most parts. There are just a few minor points left to look at.

**Minor comments**

Does the profile resulting from the fit exhibit discontinuity vertically, with jumps in concentrations across the fitted layers? How to get from the fitted quantities to the final total column value?

Please provide a concise explanation of the scattering layer's terminology or setup within FOCAL, so that the reader can understand without having to refer back to the original paper. I think this might also enhance the understanding of why the Angstrom coefficient plays a significant role in the bias correction.

**Specific comments**

45: Consider to remove "from the top of the atmosphere" for clarity.

SC7/145/357: If the scattering layer in FOCAL is assumed to be infinitely thin, how is the scattering layer be defined---usually I use the term layer to describe something between two levels. The determined scattering components, such as the Angstrom coefficient and layer height, accurately represent which specific levels (everything below the infinitesimal thin scattering level?)?

375: Is it possible to incorporate external aerosol data into FOCAL (at its current development stage)?

---

## Author Response (AR2)

Greenhouse Gas Retrievals for the CO2M mission using the FOCAL
method: First Performance Estimates
(revised version)
by S. Noël et al.

MS No.: amt-2023-194

**Authors' Response**

**Reply to comments to revised version by referee 2**

In the following, the original reviewer comments are given in *italics*, our answer in normal font.

**General comments**

1. *The refined version improves the FOCAL-CO2M retrieval algorithm's assessment and should be subject to publication. The authors have enhanced the manuscript by addressing several key issues highlighted in the review. I appreciate and thorough responses in most parts. There are just a few minor points left to look at.*

   Thank you very much for your support in improving the paper.

**Minor comments**

1. *Does the profile resulting from the fit exhibit discontinuity vertically, with jumps in concentrations across the fitted layers? How to get from the fitted quantities to the final total column value?*

   The XCO2 and XCH4 profile layers are defined such that that they contain the same number of particles. Inside a layer, all concentrations are assumed to be constant. The layers are then fitted individually such that the shape of the profiles may change. However, the amount of change is limited by the used a-priori covariance matrix. In the present case, we use matrices derived from our SLIM model, which should give a reasonable variability, therefore we do not expect large non-physical jumps between layers.

   The total column XCO2 or XCH4 is then derived from the average of the corresponding sub-columns in each layer.

   We will add this information to the paper.

2. *Please provide a concise explanation of the scattering layer's terminology or setup within FOCAL, so that the reader can understand without having to refer back to the original paper. I think this might also enhance the understanding of why the Angstrom coefficient plays a significant role in the bias correction.*

   The FOCAL scattering layer is characterised by the following quantities:

   - The vertical position of the (assumed infinitely thin) layer in terms of pressure relative to the surface pressure.
   - The optical thickness of the layer.
   - An Ångström exponent describing the wavelength dependence of the scattering.

   Scattering at this layer is assumed to be isotropic.

   All these quantities are effective, they describe the whole scattering (including e.g. Rayleigh scattering) and thus should not be interpreted as e.g. aerosol properties.

   We will add this to the paper.

**Specific comments**

1. *45: Consider to remove "from the top of the atmosphere" for clarity.*

   Will be done.

2. *SC7/145/357: If the scattering layer in FOCAL is assumed to be infinitely thin, how is the scattering layer be defined—usually I use the term layer to describe something between two levels. The determined scattering components, such as the Angstrom coefficient and layer height, accurately represent which specific levels (everything below the infinitesimal thin scattering level?)?*

   The term "layer" may indeed be misleading in the infinitesimal thin case. It should not be confused with the assumed vertical layering of the atmosphere.

   We will add the information on scattering mentioned above to clarify this.

3. *375: Is it possible to incorporate external aerosol data into FOCAL (at its current development stage)?*

   As described above, FOCAL uses effective scattering properties, therefore including e.g. external aerosol data in the FOCAL forward model is not possible.

   However, the inclusion of MAP Level 2 data (e.g. for post-processing) is already foreseen in the current software, but so far we do not have MAP Level 2 data which are consistent with our simulations.

   We will mention the latter in the paper.

**List of changes**

Changes according to the comments mentioned above have been made in the revised manuscript. The changes are marked in a corresponding version of the revised manuscript (text only since figures are unchanged) provided below.

[revised manuscript text omitted]